# Fecal Microbiota Strongly Correlates with Tissue Microbiota Composition in Colorectal Cancer but Not in Non-Small Cell Lung Cancer

**DOI:** 10.3390/ijms26020717

**Published:** 2025-01-16

**Authors:** Juan Vicente-Valor, Sofía Tesolato, Mateo Paz-Cabezas, Dulcenombre Gómez-Garre, Adriana Ortega-Hernández, Sofía de la Serna, Inmaculada Domínguez-Serrano, Jana Dziakova, Daniel Rivera, Jose-Ramón Jarabo, Ana-María Gómez-Martínez, Florentino Hernando, Antonio Torres, Pilar Iniesta

**Affiliations:** 1Department of Biochemistry and Molecular Biology, Faculty of Pharmacy, Complutense University, 28040 Madrid, Spain; juavicen@ucm.es (J.V.-V.); sofiteso@ucm.es (S.T.); 2San Carlos Health Research Institute (IdISSC), 28040 Madrid, Spain; Mateo.paz@salud.madrid.org (M.P.-C.); mgomezgarre@salud.madrid.org (D.G.-G.); a.ortega.hernandez@hotmail.com (A.O.-H.); sdlsernae@gmail.com (S.d.l.S.); inmaculadadominguezserrano@gmail.com (I.D.-S.); jana.dziakova@gmail.com (J.D.); danielriveraalonso@gmail.com (D.R.); joseramon.jarabo@salud.madrid.org (J.-R.J.); anamagom@ucm.es (A.-M.G.-M.); florhern@ucm.es (F.H.); antoniojose.torres@salud.madrid.org (A.T.); 3Biomedical Research Networking Center in Cancer (CIBERONC), Carlos III Health Institute, 28029 Madrid, Spain; 4Cardiovascular Risk Group, Microbiota Laboratory, San Carlos Hospital, 28040 Madrid, Spain; 5Department of Physiology, Faculty of Medicine, Complutense University, 28040 Madrid, Spain; 6Biomedical Research Networking Center in Cardiovascular Diseases (CIBERCV), Carlos III Health Institute, 28029 Madrid, Spain; 7Digestive Surgery Service, San Carlos Hospital, 28040 Madrid, Spain; 8Department of Surgery, Faculty of Medicine, Complutense University, 28040 Madrid, Spain; 9Thoracic Surgery Service, San Carlos Hospital, 28040 Madrid, Spain

**Keywords:** microbiota, colorectal cancer, non-small cell lung cancer

## Abstract

Microbiota could be of interest in the diagnosis of colorectal and non-small cell lung cancer (CRC and NSCLC). However, how the microbial components of tissues and feces reflect each other remains unknown. In this work, our main objective is to discover the degree of correlation between the composition of the tissue microbiota and that of the feces of patients affected by CRC and NSCLC. Specifically, we investigated tumor and non-tumor tissues from 38 recruited patients with CRC and 19 with NSCLC. DNA from samples was submitted for 16S rDNA metagenomic sequencing, followed by data analysis through the QIIME2 pipeline and further statistical processing with STATA IC16. Tumor and non-tumor tissue selected genera were highly correlated in both CRC and NSCLC (100% and 81.25%). Following this, we established tissue–feces correlations, using selected genera from a LEfSe analysis previously published. In CRC, we found a strong correlation between the taxa detected in feces and those from colorectal tissues. However, our data do not demonstrate this correlation in NSCLC. In conclusion, our findings strongly reinforce the utility of fecal microbiota as a non-invasive biomarker for CRC diagnosis, while highlighting critical distinctions for NSCLC. Furthermore, our data demonstrate that the microbiota components of tumor and non-tumor tissues are similar, with only minor differences being detected.

## 1. Introduction

Gut microbiota is considered of vital relevance in human health, and different relationships have been established between the gut microbiota composition and the risk of developing various pathologies, including different tumorigenic processes [1,2]. The gut microbiota constitutes one of the environmental factors that most influences the development of cancer [3] and several studies have reported changes in the composition of the intestinal microbiota, mainly in individuals suffering from colorectal cancer (CRC) [4].

Colorectal and non-small cell lung cancers (CRC, NSCLC) represent two of the most studied tumor diseases due to their high incidence and mortality [5]. Among the biomarkers useful in the early diagnosis of these pathologies, several bacterial genera that can be detected in feces of the affected patients have recently been described [6].

According to some authors, microbiota analyses in feces from patients with CRC are considered to reflect the microbiota composition of colorectal tissue [7]. Regarding NSCLC, gut dysbiosis in humans has been linked to inflammatory conditions in the gastrointestinal tract itself, but also in the airways, with implications in pathologies such as asthma and chronic obstructive pulmonary disease [8,9]. The existence of a gut–lung axis has been proposed to explain the influence of the composition of the intestinal microbiota on lung pathologies, including NSCLC [10,11]. The importance of the gut–lung axis has become more evident following the identification of several gut microbe-derived components and metabolites, such as short-chain fatty acids (SCFAs), as key mediators for setting the tone of the immune system [10]. Recently, it has been proposed that promising gut microbiota-based therapeutic strategies could improve the effectiveness or reduce the toxicity of immunotherapy in patients with NSCLC [11]. However, the action mechanism of the lung–gut axis is relatively unclear, and most of the microbial signals in diseases are not very precise [12].

In the present work, we report the results obtained from the analysis of microbiota in tissues from two prospective series of cancer patients. Specifically, microbiota analyses were performed in colorectal tissues from CRC patients and in lung tissues from NSCLC subjects. Data obtained have been correlated to the ones from the analysis of fecal samples from the same subjects. The latter, which were recently published, demonstrated potential clinical usefulness in CRC and NSCLC of the gut microbiota biomarkers [13]. Feces constitute an easily accessible material, but it is valuable to know if the predominant taxa detected in these samples reflect the composition of the tissue microbiota.

Our results in CRC seem to indicate a strong correlation between the taxa detected in feces and those from colorectal tissues. However, our data do not demonstrate this correlation in NSCLC. Therefore, it is important to deepen our knowledge of the mechanisms involved in the hypothetical microbiota gut–lung axis.

## 2. Results

### 2.1. Tissue Microbiota α and β Diversity Comparison Between Tumor and Non-Tumor Samples from CRC Patients

Alpha and β diversity metrics were compared among paired tumoral and non-tumoral tissue samples. Figure 1 shows a representation of α diversity metrics (Observed OTUs, Chao1 index, Shannon index, Pielou’s evenness index, and Simpson index) comparing tumor and non-tumor tissues from CRC patients. The value of α diversity metrics was very similar, and *p* values indicated no significant differences between both tissue sample types.

Regarding β diversity, Bray–Curtis and Jaccard matrices were represented using principal coordinate analysis (PCoA). Figure 2 showed no clustering between both tissue types, and statistical differences were not seen when using permutation-based multivariate analysis of variance (PERMANOVA), analysis of multivariate homogeneity of variances (PERMDISP), and analysis of similarities (ANOSIM) tests. Altogether these results indicate that the core microbiome composition was similar between both tumor and non-tumor tissues in CRC patients.

This was further confirmed by the analysis of the abundance of main phyla (Figure 3a), where similar composition is envisioned in stacked bar plots. Firmicutes were dominant in both tissue types (43.2% in tumoral and 43.9% in non-tumoral), followed by Bacteroidota (19.4% in tumor tissues and 21.9% in non-tumor samples). Proteobacteria and Actinobacteriota were slightly higher in non-tumoral tissues (18.5% vs. 18.9% and 11.9% vs. 12.6%, respectively). Overall, phylum Fusobacteriota was significantly more abundant in tumoral tissues, and this was also observed at the genus level (Figure 3b), where the only significant genus that differed between both tissue types, tumoral and non-tumoral, was *Fusobacterium* (LDA score = 4.36, *p* = 0.019).

### 2.2. Tissue Microbiota α and β Diversity Comparison Between Tumor and Non-Tumor Samples from NSCLC Patients

Microbiota diversity analysis in lung tissues also revealed the absence of significant differences between tumor and non-tumor samples. Figure 4 displays the median for the five α diversity metrics and the *p* values for their comparison.

Regarding β diversity, Bray–Curtis and Jaccard indexes did not show any clustering within tumoral or non-tumoral samples, thus indicating a similar microbial composition in both types of tissue (Figure 5).

NSCLC tissues had a similar microbial taxonomy. Analysis of main phyla (Figure 6a) revealed a shared distribution, with only slight differences: Firmicutes (35.78 vs. 32.82) and Bacteroidota (21.81 vs. 18.27) were more abundant in tumor tissues, whereas Actinobacteria (21.32 vs. 25.46) and Proteobacteria (18.12 vs. 19.77) were more abundant in non-tumor tissues. No significant differences were present at a phylum level. The LEfSe analysis (Figure 6b) revealed four differentially abundant genera between both types of tissue samples. *Cryobacterium* was more abundant in tumor tissues (LDA score = 1.88; *p* = 0.033) and *Tepidicella* (LDA score = 2.04; *p* = 0.013), *Geodermatophilus* (LDA score = 2.44; *p* = 0.045), and *Nakamurella* (LDA score = 2.57; *p* = 0.033) were increased in non-tumor tissues.

### 2.3. Fecal and Tissue Microbiota Correlations in CRC and NSCLC

We mapped microbial correlations between feces and tissues. This study was performed using selected genera from the LEfSe analysis employed in developing a gut microbiome consortium previously published by our group [6]. Concretely, 22 genera in CRC and 16 genera in NSCLC were mapped. Both in CRC and NSCLC, these analyses were performed jointly in all the tissues included in the present study (tumor and non-tumor samples), since no relevant differences had been found in any of the α and β diversity tests between both types of tissues.

Regarding the CRC group, results can be observed in Figure 7. In paired tumor and non-tumor tissue samples (Figure 7a), all 22 genera fully correlated with each other, as shown by the existence of a green diagonal. In tissue–feces mapping (Figure 7b), 20 out of 22 paired correlations were significant (red diagonal), with genus *Ruminococcaceae* and *Campylobacter* being not significant but always positively correlated (r_xy_ = 0.165, *p* = 0.323 and r_xy_ = 0.216, *p* = 0.192).

Regarding the NSCLC group, heatplots displaying correlation analyses are depicted in Figure 8. In paired tumor and non-tumor tissue samples (Figure 8a), 13 out of 16 (81.3%) genera positively correlated between samples, as was shown by the existence of a green diagonal. In this case, and in contrast to Figure 8a, the green diagonal remains more masked due to the existence of several intergender correlations. One correlation was non-significant but positive (*Salmonella*: r_xy_ = 0.464, *p* = 0.150), and two other correlations were non-significant and negative (*Clostridium sensu stricto 3*: r_xy_ = −0.060, *p* = 0.861; *Olsenella*: r_xy_ = −0.210, *p* = 0.535). In NSCLC, tissue–fecal microbiota association was characterized by the absence of significant correlations (Figure 8b).

The similarity between tumoral and non-tumoral tissues was also proved by the distribution of the 20 most abundant genera in CRC and NSCLC (Figure 9). This distribution was also more similar between tissues and feces from CRC patients than between tissues and feces from NSCLC patients. Thus, *Bacteroides* was the most abundant genus in tissue and feces from CRC subjects, while *Cutibacterium* was the first genus in NSCLC tissues and *Bacteroides* in NSCLC feces.

We also examined shared and exclusive genera OTUs by sample type. We found that feces shared around 10% more genera OTUs with tissues in CRC than in NSCLC samples (Figure 10), e.g., in CRC tissues, 73.3% and 72.9% of genera were shared with feces, whereas in NSCLC shared proportions were 61.9% and 63.6%. Moreover, the percentage of common taxa in both cancer types was higher between tumoral and non-tumoral tissues (90.2% and 81.2% in CRC and NSCLC, respectively) than in the tissue–feces pair (in CRC, 73.3% between tumoral tissues and feces and 72.9% between non-tumor tissues and feces; in NSCLC, 61.9% between tumoral tissues and feces, and 63.6% between non-tumoral tissues and feces).

### 2.4. Differences in Functional Predicted Profiles in CRC and NSLC

Despite the absence of statistically significant differences between microbial composition in tumoral and non-tumoral tissues, and the presence of significant correlations between selected bacterial genera, each type of tissue preserved a unique functional profile as predicted by the Phylogenetic Investigation of Communities by Reconstruction of Unobserved States/PICRUst2 tool, which is based on in silico predictions (Figure 11). Specifically, less counts were observed in the tumors regarding the pentose phosphate pathway, antioxidant biosynthesis (pantothenate, terpenoids), DNA repair mechanisms (base excision repair, nucleotide excision repair, mismatch repair, homologous recombination and non-homologous end joining), and immune related crosstalk (flagellar assembly and NLR signaling). These findings were consistent for both types of cancer, CRC and NSCLC (Table 1).

## 3. Discussion

In the present study, we analyzed differences in the microbiome composition of tumoral and non-tumoral tissues from patients affected by CRC or NSCLC, and mapped tissue and fecal selected genera through correlation analysis. The correlation study was performed considering the selected bacterial genera in a previous investigation, which indicated that two bacterial consortia could potentially serve as microbial biomarkers for the diagnosis of CRC and NSCLC [13].

Microbial diversity has been proposed as an indicator of disease-associated states [14]. Similarly to other authors [15,16,17], we did not find significant differences in relation to microbial diversity between tumoral and non-tumoral tissues, both in CRC and NSCLC. In fact, results about microbial diversity as an indicator of dysbiosis remain controversial according to data published on CRC [18] and NSCLC [19,20,21]. Recently, Proteobacteria has been described as dominant phyla in colorectal tissues [22,23]; however, other authors indicated that Firmicutes remained as dominant taxa in CRC samples [24], the latter results yield similar conclusions to those found in this study on CRC samples. Dominant phyla across cancer types remain controversial, although it has been suggested that there is a different dominant phylum for each tumor pathology [25]. Technical aspects could influence the microbial composition of tumor tissues [26,27]. Also, notably, several investigations address the influence of environmental factors in gut microbiome composition, but whether tissue-resident bacteria are affected by the same external factors, and their degree of implication, is still unknown [28,29].

In our series of patients affected by CRC, *Fusobacterium* and Fusobacteriota appeared, as genus and phylum, respectively, increased in tumoral tissue. *Fusobacterium nucleatum* (*Fn*) is widely reported to be involved in colorectal carcinogenicity through several mechanisms: the positive regulatory effect of cytochrome P450 enzyme CYP2J2 and matrix metalloproteinase 7 (MMP7) expression on the epithelial–mesenchymal transition has been considered as example [30]. Furthermore, the FadA adhesin and other secreted mutagens (DL-homocystine and allantoic acid) have been shown to promote genotoxicity [31]. *Fusobacterium* has also been shown to promote inflammation through NF-κB signaling, thereby reinforcing its carcinogenic properties [32,33]. Therefore, *Fn* has been largely suggested as a potential microbial biomarker for CRC screening in human studies [34]. Overall, *Fn* infection would promote tumor initiation, the proliferation, invasion, and metastasis of cancer cells, and a more inflammatory and genetically unstable background, thereby leading to CRCs with poorer prognosis and resistance to treatment.

In NSCLC, *Cryobacterium*, *Tepidicella*, *Geodermatophilus*, and *Nakamurella*, were highlighted by the LEfSe in the analysis of differences between tumoral and non-tumoral samples. Little is known about the role of *Cryobacterium* in NSCLC carcinogenesis. This Gram-positive bacterium belongs to the phylum Actinobacteria, is an aerobe heterotroph and metabolizes several sugars [35]. Its abundance has been described in gastric cancer tissues as increasing with TNM stage [36] and it takes part of a central node in one species specificity network analysis for primary tumors across 33 cancer types [37]. On the other hand, *Geodermatophilus* has been described as a gastric tissue member enriched in responders to neoadjuvant chemotherapy [38] and as a peripheral node in a species network from primary tumors [37]. In addition, *Nakamurella* has been associated with grade 3 adverse events as part of gut microbiota in locally advanced esophageal carcinoma patients treated with neoadjuvant camrelizumab and chemotherapy. No information could be found about *Tepidicella*. Notably, *Nakamurella* and *Geodermatophilus* belong to phylum Actinobacteria, which have been described as increased in airway brushes from NSCLC at early stages (I-IIIA), and as a rich source of secondary metabolites with potent anticancer and cytotoxicity properties [39].

Despite the mentioned differences highlighted by the LEfSe analysis (an algorithm designed to identify genomic features relevant to each sample), our correlation results also suggested that non-tumor tissues displayed a similar microbial composition to their paired tumor samples. Microbial profiles from both types of tissue were highly and significantly correlated in the genera analyzed in both cancer types. In CRC, 100% of the 22 genera studied, and 81.3% of the 16 genera considered in NSCLC, were significant and positively correlated among tumoral and non-tumoral tissues. Thus, we hypothesize that the stromal microbiota, rather than that of the strictly intratumor environment, participates in tumorigenesis. In fact, several articles depict that peritumoral tissue, used as reference and control in multiple studies, harbors molecular alterations that can precede future malignant lesions, known as “field cancerization” [40] or modulate the existent ones [41]. Therefore, we would be in alignment with the notion which suggests that non-tumoral adjacent tissue may represent a unique state between healthy normal tissue and tumoral tissue [42].

In the case of tissue–feces mapping in CRC, tissue microbiota correlated almost perfectly with fecal microbiota, except for the genera *Ruminococcaceae* and *Campylobacter*, which showed a positive tissue–feces correlation, although this was not significant. This finding could reflect tumor heterogeneity, which causes the outcomes to vary slightly depending on the area of the tumor being analyzed [43]. Other authors have highlighted significant differences between feces and mucosa CRC samples [23]. Instead, we focused on the similarities between CRC tissue and feces, aligning with the idea that feces may represent, to some extent, the microbial composition at the tissue level. In NSCLC, tissue–feces mapping did not yield any significant correlation, in contrast to previously published results. Specifically, a previous study found that the number of shared bacteria between lung cancer biopsy and feces was higher than those observed between normal parenchymal tissue and feces, although this difference was very subtle. In addition, *Romboutsia* and *Alistipes* showed a feces–tumor correlation [44].

Our study revealed the duality between similarities and differences between tissue types through microbial analysis. Indeed, PICRUSt2 analysis (originally developed to predict functional content of microbial communities) revealed several functional pathways, which differed between tumor and non-tumor tissues, although these differences were not significant, but consistent across both cancer types. Our results do not always correspond to the metabolic and immunological regulation described in previous works for human tumor cells [45,46,47]. It should be noted that the functional analyses whose data we now report were obtained from in silico predictions that have not been validated with external databases, due to the lack of studies similar to the ones included in our manuscript. Therefore, in order to confirm the detected differences, we emphasize the need for larger and more diverse population studies in future research. Despite the potential information withdrawn in functional profile analysis, several other mechanisms may account in microbial carcinogenesis such as quorum sensing peptides, which have been shown to increase carcinogenesis [48] and modulate the immune system [49].

A gut–lung axis has been suggested as a pathway for gut-derived diseases in the lung. However, its composition or functioning has not been fully elucidated. In this work, we demonstrate that the gut–lung axis cannot be proposed based on a reproduction of the composition of the gut microbiome into the lung microbiome. Therefore, the critical role that the intestine plays in the lung may be executed by directing immune responses outside of the local environment or through the systemic dissemination of metabolites [10,50]. Thus, physical proximity contributes to the constitution of microbial communities with common members, which is why feces and CRC tissues showed an almost complete correlation, whereas this did not occur in NSCLC.

The main novelty of the current study is the establishment of correlations between tissue microbiota and fecal microbiota in patients with CRC and NSCLC. Multiple studies provide results on the potential usefulness of stool microbiota analysis in human oncology clinics. However, there is little data that really provide contrasted knowledge regarding the reflection of tumor tissue microbiota in feces. In addition, this work provides novel data on the similarities and differences existing between tumor and non-tumor tissue microbiota in patients with CRC and NSCLC.

In conclusion, the results of the present study show that there are slight differences between the tumor and non-tumor microbiota and, at the same time, that both tissue types share microbial characteristics. These findings can be considered the main strengths of the current study; they are relevant as they support the notion that when tumor tissue is scarce, due to its small size or its consumption in diagnostic tests, the study of specific microbiota biomarkers in non-tumoral tissue may be useful in CRC and NSCLC. However, each tumor type has its own microbial profile. Furthermore, our results clearly demonstrate that the fecal microbiota reflects the microbial composition of intestinal tissue but not that of lung tissue. In fact, the bacterial taxa found to be increased in feces from cancer patients were highly correlated with tissue bacteria in CRC but not in NSCLC.

As the main limitation of this work, we consider that microbiota was evaluated through 16S rRNA sequencing instead of whole-genome sequencing, which results in a loss of depth of the data. Another limitation derives from the small number of cases analyzed, mainly in the case of NSCLC. Mainly, in the case of functional analyses, future research considering larger and more diverse population studies will be necessary.

## 4. Materials and Methods

### 4.1. Patients and Tissue Samples

Thirty-eight CRC and nineteen NSCLC samples, and their paired non-tumor tissues, were obtained from patients who had undergone potentially curative surgery at San Carlos Hospital in Madrid, Spain, between 2019 and 2024. The mean age of CRC patients (24 males and 14 females) was 71.24 ± 12 (mean ± standard deviation). Five CRCs were classified as TNM I stage, 13 as TNM II, 18 as TNM III and 2 as TNM IV. For NSCLC cases (9 males and 10 females), the mean age was 72.79 ± 7.91 (mean ± standard deviation). Fourteen NSCLCs were classified as TNM I stage, 4 as TNM III, and 1 as TNM IV. After surgical resection, all tissue samples were instantly frozen in liquid nitrogen and stored until processed at −80 °C in the Biobank of the Health Research Institute of San Carlos Hospital (IdISSC) in Madrid, using Tissue-Tek OCT as a freezing medium. Therefore, all tissues were obtained from the San Carlos Hospital Biobank (B.0000725) belonging to the San Carlos Health Research Institute (IdISSC), which is part of the national network of Biobanks, project PT2020/00074 subsidized by the Carlos III Institute of Health (ISCIII) and co-funded by the European Union through the European Regional Development Fund (ERDF). Cryostat-sectioned, hematoxylin and Eosin (H&E) stained samples from each tumor block were examined microscopically by two independent pathologists to confirm the presence of ≥80% tumor cells. Paired samples of non-tumor tissues were confirmed microscopically.

Written approval to develop this study was obtained from the Clinical Research Ethics Committee of the San Carlos Hospital in Madrid (C.I. 19/549-E_BC, 27 December 2019). In addition, written informed consent was signed by the patients prior to investigation. Cases were collected independently from gender, age of the patient, or tumor stage, and no patient had received previous chemo- or radiotherapy before diagnosis and inclusion in the study. Cancer patients previously submitted to gastrointestinal resection surgery, those affected by inflammatory diseases, and those who had received antibiotic treatment one month before surgery were excluded. Moreover, chemo- and/or radiotherapy prior to the surgery was also considered an exclusion criterion.

### 4.2. DNA Extraction and Sequencing

Prior to DNA extraction, tumor and non-tumor tissue samples from CRC and NSCLC patients were washed to remove the Tissue-Tek OCT. This was performed through the addition of 1 mL PBS 1x and subsequent centrifugation at 3000 rpm and room temperature for 15 min. Next, the total DNA extraction was performed using the QIAamp^®^ DNA Mini Kit (Qiagen, Hilden, Germany) following the manufacturer’s protocol. After the extraction, DNA concentration was measured with the Invitrogen™ Qubit™ 3 Fluorometer (Thermo Fisher Scientific, Waltham, MA, USA) and the dsDNA HS (high sensitivity) Assay (Thermo Fisher Scientific, Waltham, MA, USA).

Microbiota analysis consisted in the amplification and sequencing of the bacterial 16S ribosomal RNA (rRNA) gene, using the Ion Torrent™ technology and reagents from Life Technologies (a division of Thermo Fisher Scientific, Waltham, MA, USA). The protocol followed for library preparation and sequencing has been described previously [8]. Shortly, DNA from each sample was submitted to 2 PCR reactions in parallel, each one starting from 5 ng of input DNA, and using the reagents from the Ion 16S™ Metagenomics Kit. Reactions were set to 25 cycles, with the conditions mentioned in the manufacturer’s protocol. Two different primer sets were used to amplify 7 hypervariable regions of the gene (set 1, regions V2, V4 and V8; and set 2, regions V3, V6-7 and V9). The obtained amplicons from both reactions were quantified using the Qubit™, combined equimolarly for each sample and purified with the CleanPCR reagent (CleanNA, Waddinxveen, The Netherlands). Next, barcoded libraries were prepared with the reagents from the Ion Plus Fragment Library Kit and the Ion Xpress™ Barcode Adapters 1-96 Kit. Resulting final libraries were quantified with the Qubit™, diluted to 22 pM and pooled together. An emulsion PCR was then performed using the Ion OneTouch™ 2 System and the Ion 520™ & 530™ Kit–OT2, from which template-positive ion sphere particles (TP-ISPs) were obtained. Finally, these TP-ISPs were collected, washed, enriched using the Ion OneTouch™ES Instrument OT2, and loaded onto an Ion 530™ Chip. Sequencing was performed with the Ion S5™ System.

### 4.3. Statistical Analysis

The bioinformatic analysis of sequencing data was conducted with the Quantitative Insight Into Microbial Ecology 2 (QIIME2) pipeline [51]. Raw sequences were pre-processed as described previously [8] and assigned to OTUs using the SILVA 138 SSU Ref NR 99 identity database v138. Rarefaction was then performed to a threshold over 25K OTUs.

Five alpha diversity metrics (Observed OTUs, Chao1 richness estimate, Pielou’s evenness index, Shannon diversity index, and Simpson’s diversity index) were calculated on the rarefied OTUs profiles and compared between samples via parametric tests (*t*-test) or non-parametric tests (Kruskal–Wallis test) depending on the distribution of data. Beta diversity was also assessed through the calculation of the Jaccard and Bray–Curtis similarity indexes, using PCoA for visualization of the distances between groups. Both indexes were compared between samples through the PERMANOVA, ANOSIM, and PERMDISP tests.

Taxonomic comparison was performed via the LEfSe analysis [52] to identify the bacterial taxa with significantly different relative abundance between the compared groups. Taxa with an LDA score (log10) > 2 and *p* value < 0.05 were considered as statistically significant. Functional analysis was performed using PICRUSt2 v2.5.3 [53].

Finally, correlation analyses were performed on the relative abundance of bacterial taxa using the Spearman test. These statistical analyses were performed with STATA IC16.1 (Stata-Corp LLC, College Station, TX, USA). In all cases, statistical significance was considered at *p* value < 0.05.

## Figures and Tables

**Figure 1 ijms-26-00717-f001:**
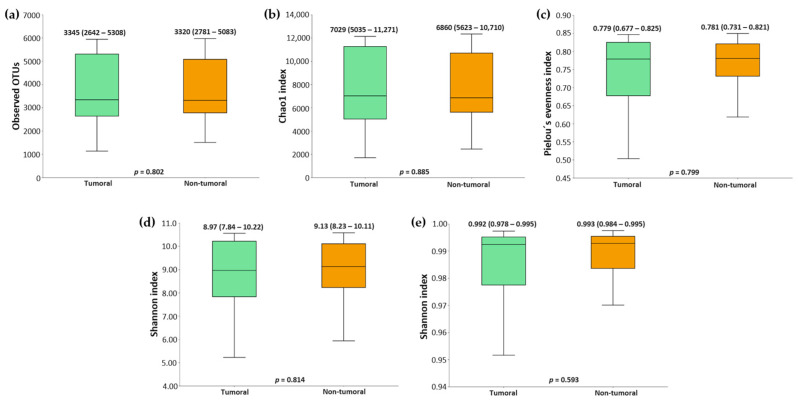
Alpha diversity comparison between tumor and non-tumor tissues from colorectal cancer (CRC) patients. (**a**) Observed OTUs. (**b**) Chao1 index. (**c**) Shannon index. (**d**) Pielou’s evenness index. (**e**) Simpson index. Median values with interquartile range and *p* values are indicated.

**Figure 2 ijms-26-00717-f002:**
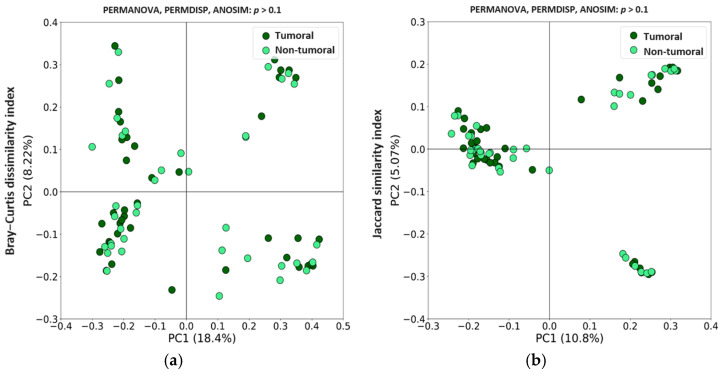
Principal coordinates analysis (PCoA) plots based on (**a**) Bray–Curtis and (**b**) Jaccard indexes for tumoral and non-tumoral tissues from colorectal cancer (CRC) patients.

**Figure 3 ijms-26-00717-f003:**
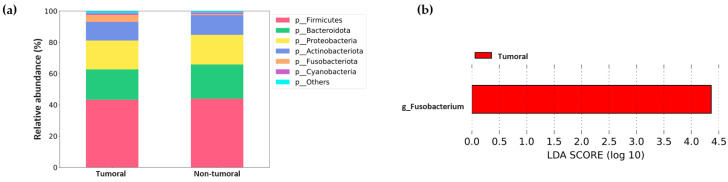
(**a**) Relative abundance of the main bacterial phyla in tumoral and non-tumoral tissues from CRC patients. (**b**) Linear discriminant analysis effect size (LEfSe) analysis showing the significant differences in bacterial genera between tumoral and non-tumoral CRC samples.

**Figure 4 ijms-26-00717-f004:**
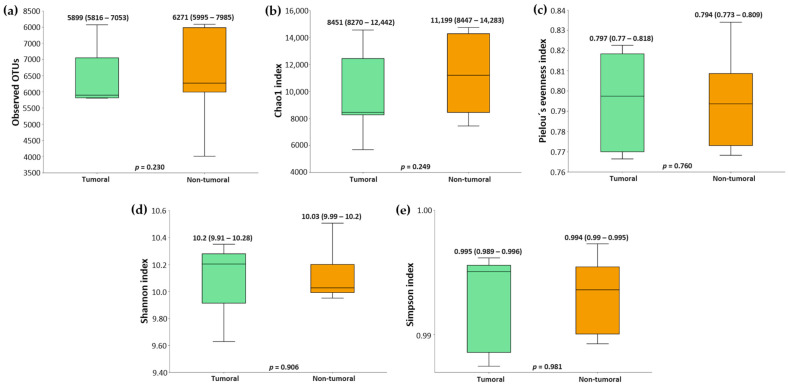
Alpha diversity comparison between tumor and non-tumor tissues from non-small cell lung cancer (NSCLC) patients. (**a**) Observed OTUs. (**b**) Chao1 index. (**c**) Shannon index. (**d**) Pielou’s evenness index. (**e**) Simpson index. Median values with interquartile range and *p* values are indicated.

**Figure 5 ijms-26-00717-f005:**
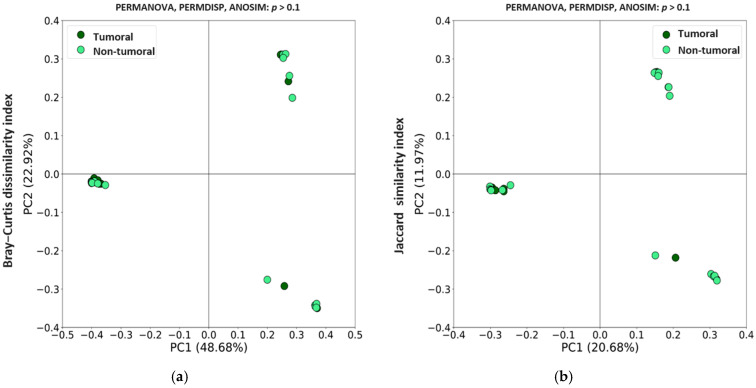
Principal coordinate analysis (PCoA) plots based on (**a**) Bray–Curtis and (**b**) Jaccard indexes for tumoral and non-tumoral tissues from non-small cell lung cancer (NSCLC) patients.

**Figure 6 ijms-26-00717-f006:**
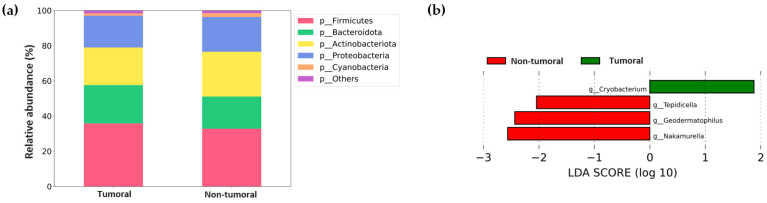
(**a**) Relative abundance of the main bacterial phyla in tumoral and non-tumoral tissues from NSCLC patients. (**b**) LEfSe analysis showing the significant differences in bacterial genera between tumoral and non-tumoral NSCLC samples.

**Figure 7 ijms-26-00717-f007:**
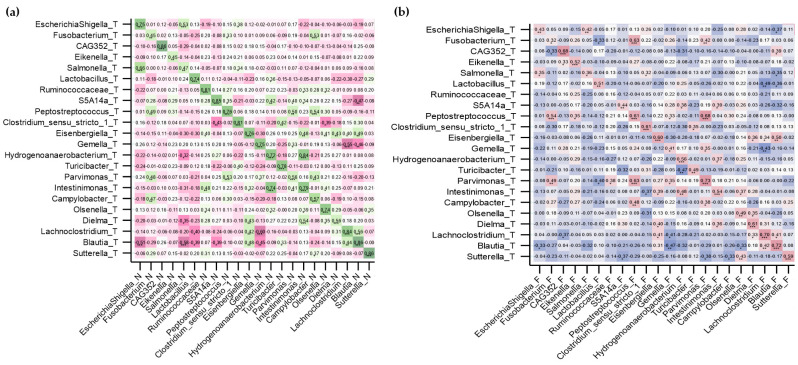
Paired correlations for selected taxa from a previously reported LEfSE analysis in CRC [6]. (**a**) Paired correlations for tumoral and non-tumoral tissues. (**b**) Paired correlations for colorectal tissues and feces. The value of the Spearman coefficient is displayed inside each square, with a star below indicating the statistical significance (* *p* < 0.05, ** *p* ≤ 0.01, *** *p* ≤ 0.001). The diagonal in both heatplots (green and red) indicates similar distribution between paired genus and paired samples.

**Figure 8 ijms-26-00717-f008:**
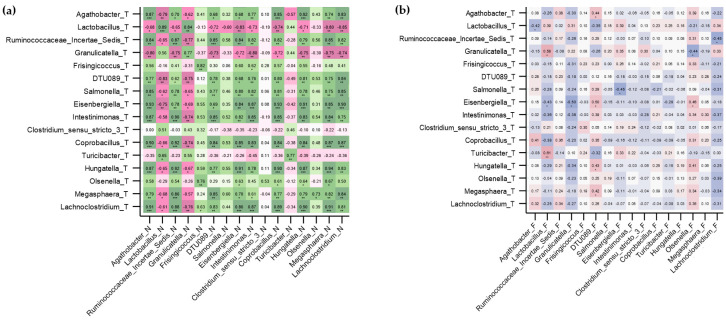
Paired correlations for selected taxa from a previously reported LEfSE analysis in NSCLC [6]. (**a**) Paired correlations for tumoral and non-tumoral tissues. (**b**) Paired correlations for lung tissues and feces. The value of the Spearman coefficient is displayed inside each square, with a star below indicating the statistical significance (* *p* < 0.05, ** *p* ≤ 0.01, *** *p* ≤ 0.001).

**Figure 9 ijms-26-00717-f009:**
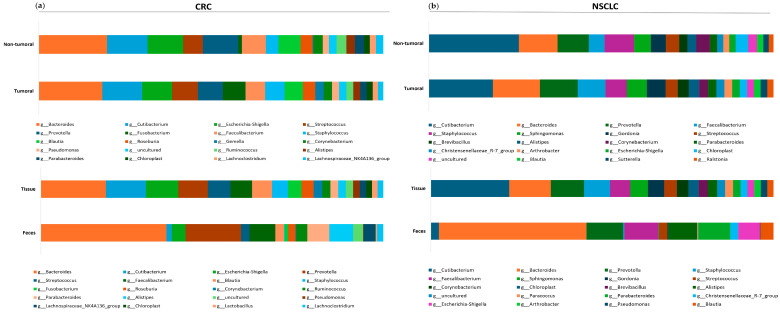
Stacked bar chart for 20 most abundant genera in tissues across all the samples in (**a**) CRC and (**b**) NSCLC.

**Figure 10 ijms-26-00717-f010:**
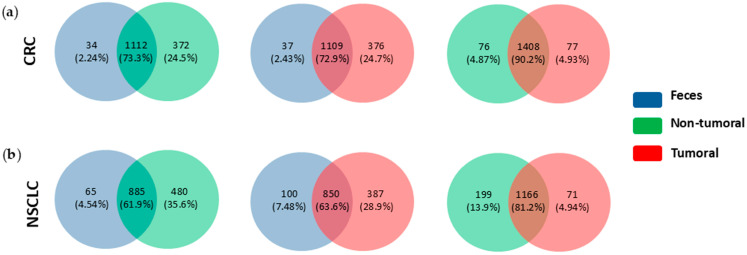
Venn diagram of unique bacterial sequences of all detected genera, across samples and cancer types: (**a**) CRC and (**b**) NSCLC.

**Figure 11 ijms-26-00717-f011:**
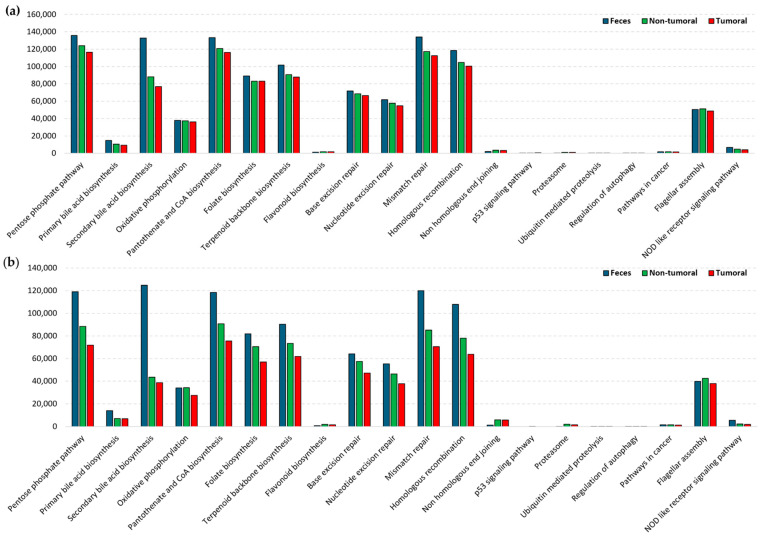
PICRUSt2 functional prediction for (**a**) CRC and (**b**) NSCLC. Bars represent counts for each pathway based on microbiota profiling.

**Table 1 ijms-26-00717-t001:** PICRUSt2 counts for selected functional pathways in CRC and NSCLC patients, for tumor and non-tumor tissues based on 16S rRNA gene amplicon sequencing on metagenomic sequencing.

Pathway	CRC		NSCLC	
	Non-Tumor	Tumor	Non-Tumor	Tumor
Pentose phosphate pathway	124,060	116,559	88,399	71,871
Pantothenate and CoA biosynthesis	120,679	116,109	90,798	75,499
Folate biosynthesis	83,097	83,136	70,500	56,893
Terpenoid backbone biosynthesis	90,661	87,912	73,355	61,911
Base excision repair	68,483	66,467	57,351	47,114
Nucleotide excision repair	57,701	54,877	46,518	37,794
Mismatch repair	117,304	112,434	85,097	70,418
Homologous recombination	104,776	100,498	77,993	63,789
Non-homologous end joining	3505	3345	57,351	47,114
Flagellar assembly	51,257	48,633	42,623	37,949
NOD like receptor signaling pathway	4732	4191	2300	2045

## Data Availability

We have deposited the raw sequence data in a public repository. Submission ID: SUB14956228. https://www.ncbi.nlm.nih.gov/bioproject/PRJNA1203234, accessed on 5 January 2025 or upon publication.

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
