# Peer review of "Fecal Microbiota Strongly Correlates with Tissue Microbiota Composition in Colorectal Cancer but Not in Non-Small Cell Lung Cancer"

_ijms, 2025, doi:10.3390/ijms26020717_

Round 1
Reviewer 1 Report
Comments and Suggestions for Authors
This study provides valuable insights into the correlation between fecal microbiota and microbiota in colorectal cancer (CRC) and non-small cell lung cancer (NSCLC) tissues. While the findings align with expectations, they contribute meaningfully to the growing understanding of the gut-CRC and gut-NSCLC axes. Below are some comments to further enhance the manuscript:
1) Cancer Staging
Could you clarify the stages of CRC and NSCLC investigated in this study? For example, is there a difference in microbial profiles or bacterial similarity between tumor and non-tumor tissues in early-onset versus invasive cancers?
2) Discussion of Microbial Metabolites
The discussion could benefit from exploring microbial metabolites involved in CRC and NSCLC/immune, such as quorum sensing peptides (QSPs) (e.g. 10.1186/s12915-022-01317-z and 10.3390/biom13020296). Did you assess the differential abundance of bacterial genera/strains associated with these metabolites in fecal and tissue samples?
3) Strengths and Limitations
A discussion of the study's strengths and limitations is currently missing and would enhance its transparency and impact. For example: a key limitation is the use of 16S rRNA sequencing instead of whole-genome sequencing, not providing strain-level resolution. Were there any biases or constraints in sample collection, handling, or cohort selection?
Author Response
1) Cancer Staging
Could you clarify the stages of CRC and NSCLC investigated in this study? For example, is there a difference in microbial profiles or bacterial similarity between tumor and non-tumor tissues in early-onset versus invasive cancers?
Response:
Two sentences detailing TNM tumor stage of CRC and NSCLC populations have been added in the “Materials and Methods” section of the revised manuscript (lines 348-349, and 350-351).
It was not possible to analyze differences between tumor and non-tumor samples considering the different TNM stages of cancers, since there were only 2 cases of advanced CRC and only 1 advanced NSCLC. As a consequence, the analysis lacks the necessary depth and quality.
2) Discussion of Microbial Metabolites
The discussion could benefit from exploring microbial metabolites involved in CRC and NSCLC/immune, such as quorum sensing peptides (QSPs) (e.g. 10.1186/s12915-022-01317-z and 10.3390/biom13020296). Did you assess the differential abundance of bacterial genera/strains associated with these metabolites in fecal and tissue samples?
Response:
Thanks for your suggestion. Our research was not aimed at finding differences between bacterial genera associated with QSP. Despite this, as 16S rRNA sequencing allows quantification of every genus, we present here data regarding differential abundance in the three genera emphasized by the proposed references: Staphylococcus, Bacillus and Enterococcus (we cannot provide exact species because rRNA 16S sequencing technology does not allow it). In CRC, Staphylococcus and Enterococcus were increased in tumor samples with regards to non-tumor samples, while in NSCLC this happened with Bacillus.
|
CRC |
|||
|
Bacteria |
Tumor vs Non-tumor |
Tumor vs Feces |
Non-tumor vs Feces |
|
Enterococcusa |
0.266 vs 0.250 |
0.266 vs 0.027 |
0.250 vs 0.027 |
|
Staphylococcusb |
3.187 vs 1.905 |
3.187 vs 0.120 |
1.905 vs 0.120 |
|
Bacillusc |
0.067 vs 0.080 |
0.080 vs 0.006 |
0.067 vs 0.006 |
|
NSCLC |
|||
|
Bacteria |
Tumor vs Non-tumor |
Tumor vs Feces |
Non-tumor vs Feces |
|
Enterococcusa |
0.189 vs 0.340 |
0.189 vs 0.015 |
0.340 vs 0.015 |
|
Staphylococcusb |
3.564 vs 4.795 |
3.564 vs 0.164 |
4.795 vs 0.164 |
|
Bacillusc |
0.337 vs 0.286 |
0.337 vs 0.012 |
0.286 vs 0.012 |
a. Including Enterococcus faecalis. b. Including: Staphylococcus aureus, Staphylococcus epidermidis c. Including Bacillus cereus, Bacillus thuringiensis
As this is a very hypothetical analysis, we have added a comment in lines 307-310, and two new references (48 and 49) in the revised version of the manuscript.
3) Strengths and Limitations
A discussion of the study's strengths and limitations is currently missing and would enhance its transparency and impact. For example: a key limitation is the use of 16S rRNA sequencing instead of whole-genome sequencing, not providing strain-level resolution. Were there any biases or constraints in sample collection, handling, or cohort selection?
Response:
The strengths and limitations of the study have been highlighted in the two last paragraphs of the Discussion section.
Also, in the Materials and Methods section (lines 351-354) that “All the cases were collected without selection in function of gender, age or tumor stage and no patient had received previous chemotherapy or radiation therapy before diagnosis and entry into this study”

Reviewer 2 Report
Comments and Suggestions for Authors
The manuscript "Fecal microbiota strongly correlates with tissue microbiota composition in Colorectal Cancer but not in Non-small Cell Lung Cancer" represents an original article on the correlation between the composition of the tissue and fecal microbiota in patients affected by colorectal or non-small cell lung carcinoma. Authors have demonstrated that 1) microbiota of tumours and surrounding tissues are similar, and 2) fecal microbiota are similar to the microbial composition of intestinal tissue but not that of lung tissue.
Considering the high incidence and serious prognosis of colorectal cancer and non-small cell lung carcinoma, the topic undoubtedly is timely and important. The content of the manuscript corresponds to the scope of the “International Journal of Molecular Sciences”, the section “Molecular Oncology” and the special edition “Molecular Mechanisms and Therapies of Colorectal Cancer: 4th Edition”, although the novelty and future implications of the findings is limited. However, up-to-dated technologies have been implemented for this research, and the data still can be useful for the global scientific community in order to show the full spectrum of scientific explorations.
Few corrections would be recommended, please:
1) In the terms “Colorectal Cancer” and “Non-small Cell Lung Cancer”, it is not necessary to use capital letters.
2) The conclusions are logical, but almost predictable. In the discussion, state clearly, please, what is the novelty of the current study and the reason to undertake it?
Finally, I would like to thank the authors for their contribution. It was a true pleasure to review this manuscript.
Author Response
1) In the terms “Colorectal Cancer” and “Non-small Cell Lung Cancer”, it is not necessary to use capital letters.
Response:
In the revised version of the manuscript, the terms “colorectal cancer” and “non-small cell lung cancer” are not in capital letters.
2) The conclusions are logical, but almost predictable. In the discussion, state clearly, please, what is the novelty of the current study and the reason to undertake it?
Response:
As the reviewer suggested, in the revised version of the manuscript, we have included a paragraph indicating the novelty of the current study and the reason to undertake it (Discussion section, lines 320-326).

Reviewer 3 Report
Comments and Suggestions for Authors
The current manuscript entitled "Fecal microbiota strongly correlates with tissue microbiota composition in Colorectal Cancer but not in Non-small Cell Lung Cancer” provides novel insights into the differential correlations between tissue and fecal microbiota in CRC and NSCLC, highlighting the strong fecal-tissue microbial mirroring in CRC but its absence in NSCLC. The findings in the abstract are not presented with sufficient emphasis on their novelty. Highlight the implications of the study for diagnosis or future research more assertively. For example: “These findings strongly reinforce the utility of fecal microbiota as a non-invasive biomarker for CRC diagnosis, while highlighting critical distinctions for NSCLC.”
In the discussion, when describing Fusobacterium, the tone is overly descriptive. Add a sentence linking these mechanisms to clinical implications. The use of technical jargon (e.g., “LEfSe analysis,” “PICRUSt2”) is not consistently explained. The functional pathway differences are described as "tenuous." Consider using a more precise term or quantifying the differences.
Moreover, I recommend performing additional experiments to enhance the impact and quality of the work. My specific comments:
· Collect longitudinal samples from the same patients before and after treatment (e.g., surgery or chemotherapy) to evaluate temporal changes in microbiota composition.
· Perform shotgun metagenomic sequencing to complement 16S rDNA results, providing more detailed functional insights into microbial community interactions and metabolic pathways.
· Increase the sample size and include diverse populations to improve statistical power and generalizability.
· Validate tissue-feces correlations through microbial culture experiments or RNA analysis to confirm active metabolic interactions.
Author Response
- The findings in the abstract are not presented with sufficient emphasis on their novelty. Highlight the implications of the study for diagnosis or future research more assertively. For example: “These findings strongly reinforce the utility of fecal microbiota as a non-invasive biomarker for CRC diagnosis, while highlighting critical distinctions for NSCLC.
Response:
The findings in the abstract are not presented with sufficient emphasis on their novelty. Highlight the implications of the study for diagnosis or future research more assertively. For example: “These findings strongly reinforce the utility of fecal microbiota as a non-invasive biomarker for CRC diagnosis, while highlighting critical distinctions for NSCLC.”
2. In the discussion, when describing Fusobacterium, the tone is overly descriptive. Add a sentence linking these mechanisms to clinical implications.
Response:
The part of the discussion regarding Fusobacterium has been modified to include more specifically its clinical implications on CRC (lines 251-256, and reference 34).
3. The use of technical jargon (e.g., “LEfSe analysis,” “PICRUSt2”) is not consistently explained. The functional pathway differences are described as "tenuous." Consider using a more precise term or quantifying the differences.
Response:
Thank you for this recommendation. We have addressed this suggestion introducing the following modifications: lines 273-274 “Despite the mentioned differences highlighted by LEfSe analysis, an algorithm designed to identify genomic features relevant to each sample, our correlation results…”; lines 300-304 “Indeed, PICRUSt2 analysis, originally developed to predict functional content of microbial communities, revealed several functional pathways which differed between tumor and non-tumor tissues, although these differences were not significant but consistent across both cancer types.
4. Moreover, I recommend performing additional experiments to enhance the impact and quality of the work. My specific comments:
- Collect longitudinal samples from the same patients before and after treatment (e.g., surgery or chemotherapy) to evaluate temporal changes in microbiota composition.
- Perform shotgun metagenomic sequencing to complement 16S rDNA results, providing more detailed functional insights into microbial community interactions and metabolic pathways.
- Increase the sample size and include diverse populations to improve statistical power and generalizability.
- Validate tissue-feces correlations through microbial culture experiments or RNA analysis to confirm active metabolic interactions.
Response:
We sincerely appreciate the reviewer's suggestions regarding additional experiments that would contribute to enriching the data provided by the manuscript. At this time, we are unable to carry them out because they are not included in the grant that supports the current study. However, we intend to apply for funding to carry them out in the near future.

Round 2
Reviewer 3 Report
Comments and Suggestions for Authors
Without incorporating either functional insights or comparisons to external datasets, the manuscript risks being perceived as incomplete, particularly in studies involving microbiota, where functionality is pivotal. The lack of validation or functional analysis reduces the potential impact and application of the findings.
By suggesting the use of online datasets and predictive methods, the authors could mitigate the resource limitations while addressing the concerns of poor functional data and enhancing the scientific value of their work. I suggest downloading and comparing their findings with data from publicly available datasets, such as those found on platforms like GEO (Gene Expression Omnibus), to provide context and validation for their results.
Author Response
Review Report Form
Open Review
(x) I would not like to sign my review report
( ) I would like to sign my review report
Quality of English Language
(x) The quality of English does not limit my understanding of the research.
( ) The English could be improved to more clearly express the research.
| Yes | Can be improved | Must be improved | Not applicable | |
| Does the introduction provide sufficient background and include all relevant references? | (x) | ( ) | ( ) | ( ) |
| Is the research design appropriate? | ( ) | ( ) | (x) | ( ) |
| Are the methods adequately described? | ( ) | (x) | ( ) | ( ) |
| Are the results clearly presented? | (x) | ( ) | ( ) | ( ) |
| Are the conclusions supported by the results? | (x) | ( ) | ( ) | ( ) |
Comments and Suggestions for Authors
Without incorporating either functional insights or comparisons to external datasets, the manuscript risks being perceived as incomplete, particularly in studies involving microbiota, where functionality is pivotal. The lack of validation or functional analysis reduces the potential impact and application of the findings.
1. By suggesting the use of online datasets and predictive methods, the authors could mitigate the resource limitations while addressing the concerns of poor functional data and enhancing the scientific value of their work. I suggest downloading and comparing their findings with data from publicly available datasets, such as those found on platforms like GEO (Gene Expression Omnibus), to provide context and validation for their results.
Response:
After browsing the suggested platform (GEO), we were not able to find metagenomic data from other studies similar to the ones included in our manuscript. The GEO expression results found in studies including both tumor and non-tumor tissues from CRC or NSCLC were based on the human genome, thus not comparable to our results which were inferred from the bacterial 16S rRNA marker gene. If the reviewer is in agreement, we have decided to remove the part of the results (lines 208 - 226 ), the discussion (lines 304 - 315) and materials and methods (line 421) regarding functional analysis, pending more robust and significant results in a larger population.
Round 3
Reviewer 3 Report
Comments and Suggestions for Authors
While it is commendable that the authors explored platforms like GEO for comparable datasets, their inability to find relevant metagenomic data highlights a limitation in the generalizability of their results. However, I suggest that rather than completely removing the sections on functional analysis, the authors should:
-
Clearly state that the functional analysis performed is based on in silico predictions.
-
Revise the Discussion to Address Limitations and Future Directions:
Acknowledge the limitation of not validating functional predictions with experimental or external datasets, and emphasize the need for larger and more diverse population studies in future research.
Author Response
-
I suggest that rather than completely removing the sections on functional analysis, the authors should:
- Clearly state that the functional analysis performed is based on in silico predictions.
- Revise the Discussion to Address Limitations and Future Directions:
Acknowledge the limitation of not validating functional predictions with experimental or external datasets, and emphasize the need for larger and more diverse population studies in future research.
Response:
Finally, the functional analyses performed in the manuscript have not been removed. As suggested by the reviewer, we have modified the results (line 209) and the discussion of the paper (lines 307-311, and 345-347), including the requested comments.